# Lysosomal Dysfunction: Connecting the Dots in the Landscape of Human Diseases

**DOI:** 10.3390/biology13010034

**Published:** 2024-01-07

**Authors:** Elisabet Uribe-Carretero, Verónica Rey, Jose Manuel Fuentes, Isaac Tamargo-Gómez

**Affiliations:** 1Departamento de Bioquímica y Biología Molecular y Genética, Facultad de Enfermería y Terapia Ocupacional, Universidad de Extremadura, 10003 Caceres, Spain; euribec@unex.es (E.U.-C.);; 2Centro de Investigación Biomédica en Red en Enfermedades Neurodegenerativa, Instituto de Salud Carlos III (CIBER-CIBERNED-ISCIII), 28029 Madrid, Spain; 3Instituto Universitario de Investigación Biosanitaria de Extremadura (INUBE), 10003 Caceres, Spain; 4Instituto de Investigación Sanitaria del Principado de Asturias (ISPA), 33011 Oviedo, Spain

**Keywords:** autophagy, lysosome, autophagosome, lysosomal storage disease, genetic mutations, therapeutic approaches

## Abstract

**Simple Summary:**

This article focuses on the impairment of lysosomes in the context of lysosomal storage disorders. Lysosomal storage disorders are a group of rare diseases with different causes united by the malfunctioning of the lysosomes. Lysosomes are intracellular vesicles, and their main function is to decompose intracellular waste in a process known as autophagy. Besides this function, lysosomes participate in a wide range of essential mechanisms aimed to keep the internal balance within our cells, which is called homeostasis. Maintaining homeostasis is a fundamental goal of all biological systems, from the simplest to the most complex, and is essential for proper functioning. Many of these disorders are originated from single-gene mutations. This provides a valuable starting point for scientists to trace the path from a single molecule to disease symptoms across the complexity of living organisms. The purpose of this paper is to provide insights from the molecular to the clinical level on this group of diseases, focusing on changes in autophagy and the latest therapeutic approaches in the field.

**Abstract:**

Lysosomes are the main organelles responsible for the degradation of macromolecules in eukaryotic cells. Beyond their fundamental role in degradation, lysosomes are involved in different physiological processes such as autophagy, nutrient sensing, and intracellular signaling. In some circumstances, lysosomal abnormalities underlie several human pathologies with different etiologies known as known as lysosomal storage disorders (LSDs). These disorders can result from deficiencies in primary lysosomal enzymes, dysfunction of lysosomal enzyme activators, alterations in modifiers that impact lysosomal function, or changes in membrane-associated proteins, among other factors. The clinical phenotype observed in affected patients hinges on the type and location of the accumulating substrate, influenced by genetic mutations and residual enzyme activity. In this context, the scientific community is dedicated to exploring potential therapeutic approaches, striving not only to extend lifespan but also to enhance the overall quality of life for individuals afflicted with LSDs. This review provides insights into lysosomal dysfunction from a molecular perspective, particularly in the context of human diseases, and highlights recent advancements and breakthroughs in this field.

## 1. Introduction

Over many years, the human lifespan has significantly increased. This fact can be attributed to several factors, including advancements in technology, the development of personalized medicine, lifestyle modifications, and the discovery of novel pharmaceutical compounds, among other contributors. Consequently, there has been an increase in the elderly population across various countries. However, the aging process is closely intertwined with the incidence of age-related diseases, including but not limited to cancer, neurodegenerative disorders, and cardiovascular diseases, which exert a significant influence on an important segment of the global demographic.

In this context, the decoding of the human genome has played a crucial role in facilitating the identification of molecular mechanisms and key contributors involved in pathologies with a genetic basis. In fact, the intricate processes associated with aging and cancer have been divided into a set of hallmark characteristics, leading to a better comprehension of these conditions [1,2]. Paradoxically, the biochemical and cellular foundations of lysosomal storage disorders (LSDs) were elucidated much earlier in scientific history, more precisely, when Christian de Duve’s research, supported by Alex B. Novikoff’s electron microscopy observations, led to the recognition of lysosomes as essential cellular catabolic organelles [3,4].

Furthermore, during this period, the biochemical abnormalities underlying some of the clinical entities described earlier were uncovered. In 1963, Pompe disease became the first disorder identified as an LSD. This milestone revealed that Pompe disease was caused by a deficiency of α-acid glucosidase, leading to the accumulation of glycogen in tissues [5]. In some particular circumstances, lysosomal dysfunction causes various human pathologies with different etiologies. In this regard, LSDs comprise a cluster of genetic metabolic diseases involving about 70 monogenic diseases attributed to an impairment of lysosomal activity [6]. These disorders may be due to, but not limited to, deficiencies in primary lysosomal enzymes, dysfunction of lysosomal enzyme activators, alterations in modifiers that influence lysosomal function, or alterations in membrane-associated proteins [7]. In fact, these type of disorders show a considerable heterogeneity, displaying different phenotypes depending on the type of substrate accumulated and its location, which may be influenced by the nature of the genetic mutation and residual enzyme activity. Therefore, LSDs are rare diseases with an estimated incidence of 1 in 4000 individuals [8]. In this regard, it is equally important to characterize the mechanisms that enable the preservation of a state of “health”. Very recently, the complex concept of health from a biologic perspective was defined by eight hallmarks, which can be categorized into three different categories: spatial compartmentalization (integrity of barriers and containment of perturbations), response to stress (repair and regeneration, homeostatic resilience, and hermetic regulation) and maintenance of homeostasis (recycling and turnover, rhythmic oscillations, and integration of circuitries) [9]. In this context, hallmark number 6, “recycling and turnover”, implies that the proper functioning of autophagy could serve as a mechanism promoting health and longevity. In fact, the autophagic process helps maintain cellular homeostasis by eliminating defective organelles and proteins as well as by coping with cellular stress. Additionally, autophagy facilitates cell renewal and an adaptive response to changes in the environment.

The term “autophagy” is derived from Greek, meaning “Self-eating”, and initially, it was identified as an adaptive response to starvation conditions in yeast. However, from an evolutionary point of view, autophagy was recognized as a conserved pathway that plays a pivotal role in maintaining cellular homeostasis across higher eukaryotes [10]. More specifically, autophagy is a cellular process responsible for breaking down dysfunctional or misfolded proteins, organelles, and various macromolecules by transporting them to the lysosomes. However, the process of autophagy can be divided into three subprocesses based on the route through which cargo is delivered to lysosomes for degradation: microautophagy, chaperone-mediated autophagy, and macroautophagy (referred to as autophagy throughout this review). In microautophagy [11], the lysosomal membrane directly engulfs cytoplasmic components. In contrast, chaperone-mediated autophagy [12] involves the selective transport of cytoplasmic proteins to the lysosome. This transport relies on the presence of a specific motif (KFERQ) in the amino acid sequence of the target protein, which is specifically recognized by the Hsc70 chaperone. Subsequently, the recognized proteins are guided to the LAMP2A protein on the lysosomal membrane, which catalyzes their translocation into the lumen for degradation. On the other hand, in macroautophagy, double-membrane vesicles called “autophagosomes” sequester different portions of cytoplasm, serving as a cellular quality-control mechanism. Finally, autophagosomes travel through the microtubular network and fuse with lysosomes, facilitating the proper degradation and recycling of their contents through the action of lysosomal hydrolases [13,14,15].

In this review, we will discuss in particular the current insights into lysosomal dysfunction from a molecular perspective, focusing on the role of autophagy and pharmacological interventions in lysosomal storage diseases.

## 2. Autophagic Process

The autophagic process is a highly complex catabolic pathway in which the coordinated functions of multiple autophagy-related (ATG) proteins are essential. At the molecular level, under nutrient-availability conditions, the ULK complex (ULK1/ULK2-ATG13-FIP200-ATG101) is inactivated due to constitutive phosphorylation by the mechanistic/mammalian target of rapamycin complex 1 known as mTORC1 (mTOR-Raptor-mLST8-Deptor-PRAS40) [16]. In addition, in these conditions, high amino acid availability activates Rag GTPases, promoting the binding of mTORC1 complex to the lysosomal membrane. mTORC1 also mediates the inactivation of the transcription factor EB (TFEB). In this regard, mTORC1 phosphorylates TFEB, promoting its cytoplasmic localization. However, during nutrient deprivation, mTORC1 is released from lysosomes, reducing its activity and thus decreasing phosphorylation levels of target proteins [17], hence decreases TFEB phosphorylation. This, coupled with calcineurin-mediated dephosphorylation, allows its translocation into the nucleus, promoting lysosomal biogenesis and gene transcription of autophagy proteins [18]. Additionally, loss of KEAP1 appears to elevate levels of TFEB transcription [19], connecting antioxidant response and autophagy. KEAP1 is an adaptor protein of a Cullin 3-based E3 ubiquitin ligase system known for its role as NRF2 (nuclear factor erythroid 2-related factor 2) main inhibitor. While it is known that KEAP1 undergoes degradation through autophagy, its specific role within this complex machinery remains unclear. However, given that a lack of KEAP1 leads to the accumulation of SQSTM1-labeled protein aggregates and vesicles with undegraded material, this would suggest its involvement in maintaining proper autophagic processes, likely in a coordinated manner with the transcriptional effect of NRF2 [20,21].

Alternatively, autophagy is also regulated by other proteins such as AMPK (AMP-activated protein kinase). In this sense, AMPK inhibits mTORC1 activity through the direct phosphorylation of Raptor and activation of TSC2 (tuberous sclerosis complex 2), which acts as a negative regulator of mTORC1 activity [22]. In addition, AMPK also activates the ULK complex, phosphorylating the ULK1 protein at different residues [23].

Upon activation, the ULK complex recruits the PI3KCIII complex, which catalyzes the conversion of phosphatidylinositol to phosphatidylinositol-3-phosphate (PI3P), creating a membrane domain enriched with PI3P and triggering the formation of the isolation membrane (IM) [24]. Furthermore, transmembrane proteins VMP1 and ATG9 participate in the events of nucleation and expansion of the IM [25]. Additionally, ATG2 supplies lipids for membrane expansion during the early stages of IM consolidation and interacts with the mammalian orthologues Atg18 and WIPI4 in the expansion process [26].

Ubiquitin-like conjugation systems (UBLs) are essential for the maturation of the IM into an autophagosome, including the ATG12 and ATG8 UBL systems. ATG12 UBL system formation is an irreversible process [27]. On the other hand, the ATG8 UBL system includes LC3 and GABARAP subfamilies. Inactive ATG8 is cleaved to its active form by the cysteine proteases ATG4 [28]. Then, it is conjugated to phosphatidylethanolamine (PE) on the membrane of the ongoing autophagosome. This process is known as ATG8 lipidation [28]. Unlike the conjugation of the ATG12 complex, the ATG8-PE conjugation can be reversed by the action of ATG4 proteases [29,30,31]. In fact, the lipidation of ATG8 proteins can occur on nonautophagic membranes that have PE, hence the importance of being a reversible process. In this sense, this delipidation is carried out by ATG4 proteases and regulated by ULK1 [32]. Although ATG16L is not required for ATG8 lipidation, its membrane binding determines the site of lipidation [33].

These conjugation systems play an additional role in cargo recognition, the regulation of autophagosome trafficking, the anchoring and fusion of autophagosomes and lysosomes, and the degradation of the inner autophagosomal membrane. In this regard, ATG8 proteins interact with specific receptors containing LC3 (LIR) or GABARAP (GIM) domains [34] such as SQSTM1, NDP52, OPTN, or NBR1.

Once formed, autophagosomes are transported to the perinuclear area, where active lysosomes are placed. This process involves motor proteins such as dynein, a minus-end-directed motor protein, or kinesin/FYCO1, a plus-end-directed motor protein. After the transport, lysosomes and autophagosomes eventually fuse, forming autolysosomes, where the cargo degradation takes place. Disruptions in the fusion process can lead to the accumulation of autophagosomes and blockage of degradation. It requires the action of different proteins, including SNAREs (soluble N-ethylmaleimide-sensitive factor attachment protein receptors) and small GTPases [35,36,37]. In this regard, proteins from the ESCRT (endosomal sorting complex required for transport) families are also essential to proper autolysosome formation. In fact, the depletion of ESCRT-0 HRS/HGS impairs autophagosome maturation and fusion with lysosomes [38,39]. A schematic overview of the autophagy process and its regulation is described in Figure 1.

Considering the highly integrated function of lysosomes and autophagosomes, it is reasonable to expect that lysosomal changes in LSDs would have an impact upon autophagy.

## 3. Lysosomal Storage Disorders Linked to Impaired Autophagy

LSDs arise from abnormal lysosomal function, leading to the accumulation of undegraded metabolites. The specific composition of these materials accumulated in lysosomes varies significantly among LSDs. Given the relevance of the lysosomal pathway in cellular homeostasis, its malfunction leads to the dysregulation of several cellular processes linked to this organelle such as lipid homeostasis, cell viability, exocytosis, membrane repair, and autophagy, among others [40]. Lysosomes are the final scenario of autophagic degradation, and the inability to clear autophagosomes results in the accumulation of undesired cargo that can further hinder cell viability. Thus, it is not surprising that defective autophagy has already been described in different LSDs [41], contributing to the development of the disease.

### 3.1. Mucopolysaccharidoses

Mucopolysaccharidoses (MPS) are a group of inherited diseases caused by mutations in genes encoding several lysosomal enzymes (including ARSB, GALNS, GLB1, GNS, GUSB, HGSNAT, HYAL1, IDS, IDUA, NAGLU, and SGSH), which are involved in the degradation of glycosaminoglycans (GAGs). However, it was described that additional alterations in other pathways (including autophagy) play a significant role in the pathogenesis of these diseases [42]. In this regard, GAGs are linear polysaccharides composed of repeating disaccharide units, which are highly sulphated. This heterogeneous group includes molecules like hyaluronic acid, heparan sulphate, dermatan sulphate, or keratan sulphate, and they are present in all human tissues. Therefore, accumulation of GAGs affects a wide range of organs and systems, and the CNS is the most commonly impacted. MPS patients exhibit a range of symptoms that differ in severity but share certain characteristic traits. These features involve facial characteristics, skeletal abnormalities, and multiorganic affectation such as heart issues, respiratory problems, and the enlargement of the liver and spleen. Thirteen different types and subtypes of MPSs have been described: MPS I (which includes three subtypes: Hurler, Hurler–Scheie, and Scheie syndromes), MPS II (also known as Hunter syndrome), MPS III (subtypes IIIA, IIIB, IIIC, and IIID), MPS IV (subtypes IVA and IVB), MPS VI, MPS VII, and MPS IX [43]. Mutations related to NCLs are listed in Appendix A [44,45,46,47,48,49,50,51,52,53,54,55,56,57,58,59,60,61,62,63,64,65,66,67,68,69,70,71,72,73,74,75,76,77,78,79,80,81,82,83,84,85,86,87,88,89,90,91,92,93,94,95,96,97,98,99,100,101,102,103,104,105,106,107,108,109,110,111,112,113,114,115,116,117,118,119,120,121,122,123,124,125,126,127,128,129,130,131].

From a mechanistic point of view, autophagy levels were recently addressed in different models of MPS, resulting in conflicting conclusions [132]. For example, impaired autophagy was described or suggested in MPS II [133], MPS III-A [134], MPS III-B [135], MPS IV-A [136], MPS VI [137], and MPS VII [138]. However, other reports also showed unaltered autophagy in MPS I and III-B [139,140] or even enhanced autophagic response in MPS I, II, and VII [141,142]. Though we cannot rule out the possibility of differential autophagy regulation in MPS subtypes, a possible explanation for this controversy could be the lack of complete, comprehensive autophagy flux analyses, which hinders the precise characterization of the autophagic pathway in these diseases, as has already been discussed in MPS III-C [143]. Interestingly, Kondo and collaborators described a new MPS-like disorder caused by a mutation in VPS33A [144], a gene that encodes for a protein that mediates autophagosome–lysosome fusion. However, this mutation (c.1492C > T; p.Arg498Trp) does not compromise the autophagy-related role of VPS33A, unveiling a new function of this protein (Figure 2: mucopolysaccharidoses).

### 3.2. Autophagy in Glycogenoses

The glycogenoses are LSDs characterized by severe autophagy defects specifically affecting skeletal and cardiac muscles. In this regard, these diseases are commonly referred to as autophagic vacuolar myopathies (AVMs), although neuropathological manifestations are also observed in some particular cases [145]. Conditions leading to AVM are associated with genetic mutations affecting genes responsible for glycogen hydrolysis, lysosome acidification, and the maturation and fusion of autophagosomes with lysosomes [146]. Mutations related to glycogenoses are listed in Appendix A [147,148,149,150,151,152,153,154,155,156,157,158,159,160,161,162,163,164,165,166,167,168,169,170,171,172,173,174,175,176].

#### 3.2.1. Pompe Disease

Pompe disease is a rare disorder caused by autosomal recessive mutations in the GAA gene that result in the accumulation of glycogen within lysosomes [177,178]. It was the first recognized LSD [5] and is also known as glycogen storage disease type II (GSDII). Different studies established an association between specific GAA single-nucleotide polymorphisms (SNPs) and Pompe disease, including rs140826989, rs121907938, rs121907945, rs121907936, rs147804176, rs1555600061, rs1800312, and rs200856561, among others [155,170,173]. Clinically, Pompe disease includes two main groups: infantile-onset Pompe disease (IOPD) and late-onset Pompe disease (LOPD) [179]. IOPD patients typically exhibit severe symptoms within the first few months of life, including weakness, developmental delay, feeding difficulties, hypotonia, macroglossia, and hypertrophic cardiomyopathy. These patients experience multisystem glycogen accumulation with less than 1% of normal GAA enzyme function. Without treatment, their life expectancy is typically less than two years [180].

On the other hand, LOPD patients may present symptoms anywhere from childhood to adulthood, with worse prognosis when symptoms manifest at an earlier age. These patients generally retain 1–20% of normal GAA function. However, abnormal glycogen build-up in the respiratory system can lead to respiratory failure, necessitating mechanical ventilation in a significant percentage of patients [181]. From a molecular perspective, the glycogen accumulation interferes with cellular processes such as metabolism or autophagy. In this context, the accumulation of autophagy substrates and autophagosomes was demonstrated by using knockout mice models of GAA [182,183]. In fact, components of the autophagy system, including BECN1, GABARAP, LC3, and ATG7, are increased; however, this excess is associated with functional autophagy deficiency, as the machinery is unable to degrade them. Furthermore, in primary cells derived from KO mice, lysosomal acidification is inefficient, leading to a blockade of autophagy and an impairment of mitochondrial function, which are associated with this lysosomal disorder [184,185].

#### 3.2.2. Danon Disease

Danon disease was the first LSD in which an association with autophagy involvement was reported. Danon disease it is characterized by a deficiency in LAMP2 (lysosomal-associated membrane protein 2). The disease is inherited as an X-linked trait and is exceptionally rare [186]. From a phenotypic point of view, Danon disease is characterized by severe hypertrophic cardiomyopathy, heart failure, muscle weakness, retinopathy, and different degrees of mental retardation specifically in male patients. However, in female patients, it is described as a “milder phenotype” that is mostly limited to cardiac abnormalities.

According to its relation with the autophagic process, the accumulation of large LC3-positive membrane-bound structures is a very common feature in several tissues, especially in muscular tissues. In this context, an increase in lipofuscin accumulation and myofibrillar disruption are present in cardiac muscle too.

From a molecular point of view, most Danon patients carry mutations that result in LAMP2 (lysosomal-associated membrane protein 2) loss of function. In fact, Danon disease is also known as “glycogen storage disease due to LAMP2 deficiency”. These alterations cause an important failure in autophagosomal–lysosomal fusion accompanied by an excessive accumulation of autophagosomes and partial modifications in a subset of lysosomal enzymes and p62 aggregates. In this regard, by using muscle biopsies from patients, a correlation was reported between a blockage in autophagy flux and disease severity. In addition, mice deficient in Lamp2 mimic Danon disease in humans, showing an accumulation of autophagosomes, which severely affects cardiac contractile function [187].

Although the most Danon-affected people present a deficiency of all three LAMP-2 isoforms, the pathogenesis and clinical manifestations are attributed to the specific deficiency of LAMP-2B, the expression of which is abundant in the heart, muscle, and brain. Indeed, only a defect in LAMP-2B is sufficient to cause most of the disease manifestations [188]. Furthermore, these patients present defects also in mitochondrial clearance (mitophagy), which shows that it is not only bulk autophagy that is altered in this type of disease [189] (Figure 3).

### 3.3. Autophagic Process in Sphingolipidoses

Sphingolipids represent a major category of lipids in the nervous system and play an important role in neural development and functionality [190]. Their metabolism is tightly regulated through a multistep degradation process that relies on several lysosomal hydrolases [191]. In this regard, sphingolipidoses represent a broad group of inherited disorders related to sphingolipid metabolism that frequently affect the nervous system. Specifically, sphingolipidoses are a subset of LSDs characterized by the accumulation of partially or completely undegraded sphingolipids. These alterations are observed mainly in the pediatric population, manifesting in neurodegeneration that leads to psychomotor retardation and myoclonus due to widespread and progressive damage to neurons. In some cases, these conditions may cause weakness and spasticity due to the involvement of white-matter tracts. The genetic changes underlying these disorders are diverse and can lead to the accumulation of substances such as sphingomyelin, glycolipids, glucocerebrosides, gangliosides, unesterified cholesterol, and sulfatide compounds, among others [192]. For these reasons, it is not surprising that several disturbances in autophagy have been reported in sphingolipidoses. For example, the addition of glycosphingolipids to cells, even by a simple method such as supplementation in a culture medium, triggers autophagy, prevents the clearance of autophagosomes, and leads to their accumulation [193]. Diseases in this category include Niemann–Pick, Gaucher, and Fabry diseases as well as mucolipidoses, and GM1/2 gangliosidoses such as Tay–Sachs and Sandhoff diseases [194]. Mutations related to sphingolipidoses are listed in Appendix A [195,196,197,198,199,200,201,202,203,204,205,206,207,208,209,210,211,212,213,214,215].

#### 3.3.1. Gaucher Disease

Gaucher disease (GD) is one of the most common lysosomal storage disorders and occurs in up to 1 in 40,000 live births in the general population. GD disease includes three clinical phenotypes: GD types I, II, and III [216]. GD type I is considered a systemic disorder, with no neurological involvement. This type represents more than 90% of GD cases; the common symptoms are splenomegaly, hepatomegaly, thrombocytopenia, coagulation abnormalities, anemia, bone disease, and bone marrow infiltration by storage cells. The age of onset is highly variable and can start at any time from childhood to 70 years old. However, most patients are diagnosed before the age of 20 [217].

Several studies pointed to a genetic association between the p.N370S allele, both homozygous and heterozygous states, and the development of GD type I [218]. In contrast, GD type II often manifests with severe symptoms within the first six months of life. Patients with this form of the disease experience neurological manifestations, brainstem involvement, developmental delays, cachexia, respiratory distress, pneumonia, skin issues, and other symptoms. The prognosis for individuals with GD type II is very poor, as infants rapidly deteriorate, and unfortunately, most do not survive beyond 2 years of age [219]. GD type III (commonly known as chronic neuropathic form) has a more insidious course. The manifestations are developmental delay; strabismus and supranuclear gaze palsy; progressive dementia; myoclonus; corneal opacities; and cardiovascular calcification. From a genetic point of view, it is known that individuals who are homozygous for the p.L444P allele probably develop type II or III [220,221]. In this regard, glucosylceramide and glucosylsphingosine are the primary glycosphingolipids that accumulate in this disease, primarily in macrophages known as “Gaucher cells”, which are found in the liver, spleen, lungs, and central nervous system. From a molecular point of view, these accumulations result from mutations in the GBA1 gene, which encodes glucocerebrosidase [222].

In GD, there is an impairment of the autophagic process. In fact, it was described as an accumulation of autophagosomes in different in vitro and in vivo models. For example, using induced pluripotent stem cells (iPSCs) derived from Gaucher patient cells that were reprogrammed into neurons, the authors observed an increase in the number of autophagosomes and the amount of autophagic markers such as LC3-II and p62, which appear exclusively in neuropathic cells. Specifically, affected neuropathic cells display blocked autophagic flux with reduced autophagosomal clearance and decreased levels of LAMP1 and TFEB. This indicates a lysosomal dysfunction and is a probable cause of neurodegeneration in this pathology. The overexpression of TFEB in combination with recombinant glucocerebrosidase treatment ameliorates these alterations [223].

Furthermore, a mouse model that phenocopies GD mediated by mutations in both glucocerebrosidase (V394L) and C-saposin deficiencies shows accumulation of p62/SQSTM1 in neurons and astrocytes along with sequestration of undigested materials within axonal vesicles. These observations indicate that autolysosomal cargo degradation is impaired in cells affected by GD [224].

In summary, autophagy dysfunction is present in various model systems of GD, but the underlying mechanisms are still unclear.

#### 3.3.2. Niemann–Pick Type C Disease (NPC)

Niemann–Pick type C (NPC) disease is a genetic autosomal recessive lysosomal storage disorder caused by mutations in either NPC1 (95% of cases) or NPC2 (5% of cases). These genes encode proteins involved in the intracellular trafficking of lipids and cholesterol [225]. Mutations in these genes result in the accumulation of unesterified cholesterol in the liver, spleen, and brain, which, in turn, disrupts lipid transport. In fact, these alterations cause a disruption that leads to the loss of Purkinje cells in the cerebellum and degeneration of other components of the central nervous system [226].

From a clinical perspective, NPC is typically a disease with juvenile or later onset, and the rate of progression inversely correlates with the age of onset. Common symptoms of NPC include ataxia, splenomegaly, hepatomegaly, hypotonia, severe liver disease, respiratory infections, and abnormal eye movements [227].

In this context, there is an alteration of the autophagic mechanism in NPC, as an accumulation of autophagosomes in skin fibroblasts from NPC patients is described. In the molecular landscape of this disease, this accumulation is partially due to the function of BECN1 and LC3B. In wild-type fibroblasts, their levels increase when exposed to U18666A, a small molecule used to induce NPC-like lipid trafficking defects. Moreover, NPC exhibits a blocked autophagic flux due to impaired autophagosome maturation [228] and specific defects in mitophagy [229].

Therefore, autophagy is significantly disrupted in NPC. This disruption interferes in the maintenance of cellular and tissue homeostasis, contributing to the pathological changes observed in NPC patients. The accumulation of autophagosomes, their impaired maturation, and the defective mitochondrial function all contribute to the disease’s progression, affecting cellular and tissue health.

#### 3.3.3. Fabry Disease

Fabry disease (FD) is an X-linked LSD characterized by mutations in the GLA gene. This gene encodes the lysosomal enzyme α-galactosidase A (α-Gal A). Deficiency of α-Gal A causes an accumulation of globotriaosylceramide (Gb3) in organs, including the heart, kidneys, brain, and eyes, among others [230]. FD is among the more frequent LSDs after GD [231]. Due to its X-linked inheritance pattern, prevalence in males is higher than in females. In males, there are two subtypes: the classic presentation with a severe phenotype and the nonclassic with a less severe phenotype [232]. The classical phenotype of FD has an incidence of 1:22,000 to 1:40,000 in males, while the nonclassical form oscillates from 1:1000 to 1:3000 in males and 1:6000 to 1:40,000 in females. In this regard, patients with the classic phenotype have less than 1% of normal α-Gal A activity and tend to develop complications and symptoms earlier in life. Patients with the nonclassic subtype have a milder form of the disease, with higher α-Gal A activity [233]. In contrast, although females are typically asymptomatic, a small percentage of them can exhibit a milder pattern of the disease due to continued secretion of α-Gal A from their other X chromosome.

Clinical features of FD include auditory disturbances, renal disease, cerebrovascular disease, cardiac hypertrophy, arrhythmia, angiokeratoma (skin lesions), and excessive sweating [234]. From a molecular perspective, over 900 mutations have been identified in association with FD; D313Y, E66Q, and A143T are among the most common associated mutations [235,236].

In this scenario, taking into account the rest of the sphingolipidoses, in FD there is an increased basal expression of the autophagosome marker LC3-II, as observed in cultured cells from FD patients compared to wild-type cells. Furthermore, human podocytes derived from patients with FD have increased expression of GABARAP and BECN1 along with an inhibition of mTOR. However, an impairment in the autophagy flux in FD is described. According to this alteration, an increased staining of p62 and ubiquitin was observed in renal tissues and cultured fibroblasts from FD patients [237]. Moreover, the accumulation of autophagy substrates, autophagosomes, and lysosomes was proved using an α-Gal A-deficient mouse model [238]. These changes suggest that autophagy dysfunction could contribute to the progression of neuropathological changes in FD [239] (Figure 4).

### 3.4. Autophagy Pathway in Neuronal Ceroid Lipofuscinoses

Neuronal ceroid lipofuscinoses (NCLs) are one of the most prevalent causes of neurodegeneration in children. NCLs typically present with symptoms such as blindness, seizures, progressive cognitive decline, and motor impairment. NCLs exhibit both genetic and phenotypic diversity. The juvenile onset NCL (JNCL), also known as Batten disease, is the most common presentation and is caused by mutations in the CLN3 gene [240]. However, NCLs could manifest as different subtypes, each named according to the specific mutations in specific ceroid lipofuscinosis neuronal (CLN) genes such as CLN1, CLN2, CLN4, CLN5, CLN6, CLN7, CLN8, CLN10, CLN11, CLN12, or CLN13 [241]. Mutations related to NCLs are listed in Appendix A [242,243,244,245,246,247,248,249,250,251,252,253,254,255,256,257,258,259,260,261,262,263].

The predicted functions of CLN proteins are diverse, with some acting as lysosomal enzymes, while others are thought to regulate intracellular trafficking or membrane transport. Unfortunately, the precise cellular roles of most CLN proteins remain a mystery, prompting the utilization of a wide variety of model systems in NCL research. However, these diseases share the common features of autophagosome accumulation, dysfunctional mitochondria, and alterations in autophagy-related pathways, providing important insights into their pathogenesis [264,265].

Batten disease is an autosomal recessive disorder typically affecting children between the ages of 5 and 10 caused by mutations in CLN3. A frequently observed defect in CLN3 is the homozygous deletion of 966 base pairs, encompassing exons 7 and 8, resulting in a premature stop codon in exon 9 [266]. In this context, the CLN3 protein is located in many cellular compartments, including the endo-lysosomal pathway and the Golgi complex. However, although its exact function is not fully understood, it has been linked to intracellular trafficking through interactions with Rab7A and protein secretion processes [267,268].

In this context, several mammalian models of NCLIII disease have shown reduced trafficking and levels of lysosomal enzymes. Early-stage defects in the autophagy pathway have been observed in both mouse and human cellular models with CLN3 mutations, leading to the accumulation of autophagosomes and autolysosomes. In particular, Cln3Dex7/8 knock-in mice and Cln3Dex7/8 cerebellar cells exhibit increased levels of LC3-II, downregulation of mTORC1, accumulation of autophagosomes, and impaired maturation as well as impaired turnover of ATP synthase subunit C [269,270].

On the other hand, the NCL X subtype (characterized by mutations in the Cathepsin D gene) also presents an increase in the number of autophagosomes and an accumulation of malfunctioning mitochondria due to an impairment in the autophagy flux [271] (Figure 5).

### 3.5. Autophagy in Glycoproteinoses

Glycoproteinoses form a category of lysosomal diseases resulting from deficiencies in the catabolism of glycoproteins. In this regard, glycoproteins are prevalent components found in cells and on cell surfaces. These genetic disorders follow an autosomal recessive inheritance pattern. Glycoproteinoses share the deficit of specific lysosomal enzymes that are crucial for the systematic breakdown of glycoprotein glucids [272]. Pathogenic sequence variants in genes encoding these enzymes lead to glycoproteinoses. The oligosaccharide composition serves as an indicator of glycoproteinosis, potentially offering insights into a specific diagnosis [273]. In this context, there are some deficiencies encompassed in this type of pathology, such as fucosidosis, galactosialidosis, Schindler’s disease, α-Mannosidosis, or β-Mannosidosis, among others [274]. In many of these disorders, the accumulation of undigested material induces vacuolization in cells, such as peripheral blood cells and fibroblasts. This accumulation can also have pleiotropic effects on cellular functions, including synaptic release, exocytosis, and autophagy [273]. Mutations related to glycoproteinoses are listed in Appendix A [275,276,277,278,279,280,281,282,283,284,285,286,287,288,289].

#### 3.5.1. α-Mannosidosis

α-mannosidosis, is an uncommon LSD (1:500.000) and follows an autosomal recessive inheritance pattern. Typically, it is unnoticeable at birth and displays symptoms progressively [290]. Clinical features of this disease include cognitive developmental delay, hearing loss, skeletal deformities, central nervous system involvement, and immunodeficiencies. Traditionally, α-mannosidosis is classified into two categories, according to its severity. However, wider classification includes three clinical types differentiated by the age of onset, speed of progression, and presence/absence of skeletal abnormalities. Type 1 includes mild presentation, an age of onset after 10 years old, no skeletal implication, and very slow progression. Type 2 is the moderate form, in which symptoms are identified before 10 years old, including skeletal abnormalities, with a slow progression that leads to ataxia around the age of 30. Type 3 is the most severe presentation and is immediately recognizable, exhibiting skeletal abnormalities. It progresses fast, culminating in early death due to central nervous system involvement or myopathy. The most frequent presentation is Type 2 [291,292,293].

From a genetic point of view, α-mannosidosis is originated by MAN2B1 gene mutations, generating an aberrant enzyme (α-mannosidase) that is unable to break down mannose-containing oligosaccharides [294]. Consequently, these accumulate within lysosomes, triggering cell malfunction and eventual cell death. The build-up of oligosaccharides and ensuing cell death contribute to tissue and organ damage, resulting in the distinctive features observed in α-mannosidosis [295].

#### 3.5.2. β-Mannosidosis

β-mannosidosis is an autosomal recessive LSD that occurs as a result of a dysfunctional β-mannosidase enzyme. While well-known and relatively common in goats and other types of cattle, it is exceptionally rare in humans. For this reason, the characterization of this pathology in a human context poses significant challenges due to its infrequency and limited occurrence in the human population [296,297].

The signs and symptoms of β-mannosidosis present a wide spectrum of severity, and onset can occur from infancy to adulthood. In fact, there is no clear genotype/phenotype correlation. Common symptoms include intellectual disability, and some may experience delayed motor development and seizures. Behavioral problems, such as hyperactivity, impulsivity, aggressiveness, or a tendency to depression; respiratory and ear infections; hearing loss; speech impairment; swallowing difficulties; and hypotonia, are additional challenges faced by those with β-mannosidosis and are also common among patients who may exhibit introverted behavior [298,299]. Distinctive facial features and the presence of small angiokeratomas, formed by clusters of enlarged blood vessels, are also observed in individuals with this condition.

From a molecular point of view, β-mannosidosis is caused by alterations in the MANBA gene [300]. These disrupt the function of the β-mannosidase enzyme, which serves as the final exoglycosidase in the degradation of N-linked oligosaccharides of glycoproteins, removing β-linked mannose residues [301]. Affected individuals exhibit a significant reduction in β-mannosidase activity, and this leads to the lysosomal accumulation of disaccharides.

## 4. Pharmacological Modulation

Although LSDs are a very heterogeneous group of diseases, in recent years, patients that suffer from these pathologies have experienced a notable extension in life expectancy. This milestone is possible thanks to different factors, including the exploration of innovative pharmaceutical compounds, among other contributory elements [8,302].

In the treatment of LSDs, various approaches are available. However, most LSDs are a result of a deficiency in lysosomal hydrolases. In this context, the primary approach is based on replacing the deficient enzyme activity with a fully functional wild-type enzyme. Among these strategies, one of the most extensively studied methods is enzyme replacement therapy (ERT). ERT operates on the premise that lysosomal enzymes can be taken up by cells and effectively delivered to lysosomes through the mannose-6-phosphate pathway [303].

ERT is now the established treatment for numerous LSDs. Common disorders such as Pompe disease, Gaucher disease, Fabry disease, or mucopolysaccharidoses are successfully treated using ERT. However, the use of these pharmacological therapies presents potential issues related to immune system responses [303]. On the one hand, the development of antibodies against therapeutic enzymes can impact treatment effectiveness or trigger immune-mediated severe adverse reactions, necessitating complex and expensive protocols to induce immune tolerance [304]. On the other hand, there is a problem with the specific distribution of recombinant enzymes because these enzymes cannot cross the blood–brain barrier, which is clinically significant because many LSDs cause neurological symptoms [305].

To address these challenges, there is a current emphasis on the development of second-generation recombinant enzymes with improved pharmacodynamics and targeting properties. For example, in the context of Pompe disease, the generation of avalglucosidase-alpha, a glycoengineered GAA, is currently being studied [306].

Other approaches include the use of pharmacological chaperone therapy (PCT) [307]. These therapies are based on the use of molecular chaperones to stabilize the conformation of unstable or misfolded proteins resulting from the genetic mutations that cause LSDs. From a clinical perspective, this process helps prevent the degradation of these proteins, enabling them to carry out their enzymatic functions. However, this process presents only one approved therapy, known as migalastat, for the treatment of Fabry disease [308]. Moreover, ambroxol is currently undergoing a phase 2 clinical trial as a potential therapeutic option for GD [309].

The main problem of this strategy is that the effectiveness of the PCT approach is limited to mutations affecting the stability of the enzyme and may not be suitable for cases where the mutation targets the catalytic site. In this regard, there is a demonstration of a potential synergy between PCT and ERT [309]. This synergy was initially observed both in vitro and in vivo using Pompe disease patient cells and in a murine model of the disease, and it is currently under clinical development [310,311]. The main advantage of this approach is that it directs the drug’s effect toward the recombinant enzyme used in ERT rather than the endogenous mutant enzyme. Consequently, the effect of chaperones in this approach is mutation-independent and can theoretically be extended to all ERT-treated patients.

Substrate reduction therapy (SRT) has emerged as a potential treatment approach for addressing neurological manifestations [312,313]. These small molecules are designed to mitigate the build-up of undegraded substances by limiting either their synthesis or the synthesis of their precursors. Two notable medications, miglustat and eliglustat, have received approval for the treatment of Gaucher disease, serving as second- and first-line therapies, respectively [314,315]. However, it is worth noting that miglustat has not demonstrated superior efficacy compared to ERT. Additionally, eliglustat is incapable of crossing the blood–brain barrier, making it unsuitable for treating forms of the disease that involve the nervous system [316].

Remarkably, miglustat has also gained approval from the European Medicines Agency for the treatment of Niemann–Pick type C disease due to its significant impact on reducing the progression of neurological disease [226]. In fact, other substrate-reducing compounds are currently in clinical development or are approved for the treatment of Gaucher and Fabry diseases, including, for example, venglustat [233].

In addition, another type of approach implies the introduction of nucleic acids into cells using either adeno-associated viral (AAV) vectors in vivo or lentiviral vectors ex vivo to benefit patients. This gene therapy, in the context of LSDs, typically means delivering a functional copy of the defective gene [317]. LSDs are ideal candidates for gene therapy because they are generally well-characterized monogenic disorders, meaning they are caused by mutations in a single gene. In these cases, the defective gene has been identified and there is availability of animal models already developed for preclinical testing. In AAV-mediated in vivo gene transfer, a vector is administrated through systemic or intraparenchymal injection; this vector carries the therapeutic gene under the control of ubiquitous or organ-specific promoters [318]. Ex vivo gene therapy, on the other hand, involves the correction of a patient’s cells, such as hematopoietic stem cells, outside the body. The genetic modification of these cells is performed in vitro, and then the cells are reimplanted into the patient. For some LSDs, the newly produced enzyme is released into the bloodstream, available to be used by cells throughout the body. This process, known as “cross-correction,” enables specific organs like the liver to act as enzyme factories, producing the protein and delivering it to other organs through the bloodstream [319]. Importantly, the estimated enzyme replacement percentage needed to reach the effective clearance of the accumulating substrate is relatively low (1–10%), though this can vary depending of the specific disease. Therefore, fully restored levels of the functional enzyme may not be necessary to alleviate the symptomatology of the disease [320].

However, LSDs caused by defective nonsecreted membrane-bound proteins present a greater challenge because cross-correction is not possible. Additionally, like the ERT strategy, there is the issue of the need for sustained expression of the therapeutic enzyme and the addressing of neuropathological issues [321,322].

Another strategy under investigation is the use of antisense oligonucleotides, which can restore normal splicing of transcripts. This approach shows promising results for late-onset Pompe disease patients, particularly in those with the C.32-13T>C mutation. This mutation causes aberrant splicing and exon 2 skipping, leading to reduced mRNA synthesis. This is a highly prevalent variant among affected individuals; consequently, it could benefit a large number of patients [323]. Table 1 includes approved and ongoing therapeutic approaches for the diseases included in this review.

On the other hand, autophagy has been harnessed for therapeutic advantage in several transgenic models that phenocopy different human diseases, encompassing various neurodegenerative disorders, specific liver diseases, or myopathies, among others [324]. In this regard, the initiation of autophagy not only mitigates the disease phenotypes but also, in numerous cases, contributes to increased longevity of the organism [325,326]. For this reason, autophagy could be a viable target in lysosomal storage disorders as a treatment strategy [146]. However, the most applicable disease contexts are likely to be within LSDs where lysosomal hydrolytic function is not severely compromised, for example, due to direct mutation of an enzyme. In fact, in this type of scenario, the restoration of autophagic flux could allow the digestion of the accumulated autophagic load, acting as an interesting therapeutic approach (Table 2).

**Table 1 biology-13-00034-t001:** Overview of several therapeutic approaches in lysosomal storage disorders. ^(^*^)^ Approved. References include published articles and ongoing clinical trials registered with https://clinicaltrials.gov/.

Type of LSD	Disease	Gene	Affected Enzyme	Product	Approach	Reference	Phase
Mucopolysaccharidoses	MPS I	IDUA	α-L-iduronidase	Laronidase	Enzyme replacement therapy	[327]	Appd ^(^*^)^
				RGX-111	In vivo gene therapy vectors	NCT03580083	I/II
				SB913	In vivo gene therapy vectors	NCT03041324 NCT04628871	I/II
				ISP-001	Ex vivo gene therapy	NCT04284254	I/II
				IDUA LV	Ex vivo gene therapy	NCT03488394	I/II
	MPS II	IDS	Iduronate-2-sulfatase	Idursulfasa	Enzyme replacement therapy	[328]	Appd
				AGT-182	Enzyme replacement therapy	NCT02262338	I
				JR-141	Enzyme replacement therapy	[329]	Appd
				RGX-121	In vivo gene therapy vectors	NCT0457190	I/II
				SBFIX	In vivo gene therapy vectors	NCT04628871	I/II
	MPS IIIA	SGSH	Heparan-sulfatase	rhSGSH	Enzyme replacement therapy	NCT01155778	I/II
				HGT-1410	Enzyme replacement therapy	NCT02060526	I/II
				SOBI003	Enzyme replacement therapy	NCT03423186	I/II
				SAF-301	In vivo gene therapy vectors	NCT01474343	I/II
				LYS-SAF302	In vivo gene therapy vectors	NCT03612869	II/III
				ABO-102	In vivo gene therapy vectors	NCT04088734	I/II
	MPS IIIB	NAGLU	Nac glucosaminidase	rhNAGLU-IGF2	Enzyme replacement therapy	NCT02754076 NCT03784287	I/II
				AAV5-hNAGLU	In vivo gene therapy vectors	NCT03300453	I/II
				ABO-101	In vivo gene therapy vectors	NCT03315182	I/II
	MPS IVA	GALNS	galNAc6S sulfatase	Vimizim	Enzyme replacement therapy	NCT01415427	III
	MPS VI	ARSB	Arylsulfatase B	Odiparcil	Substrate reduction therapy	NCT03370653	II
				Galsufase	Enzyme replacement therapy	[329]	Appd
Glycogenoses	Pompe disease	GAA	α-glycosidase	Alglucosidase *α*	Enzyme replacement therapy	[330]	Appd
				Avalglucosidase *α*	Enzyme replacement therapy	[306]	Appd
				VAL-1221	Enzyme replacement therapy	NCT02898753	I/II
				ATB200	Enzyme replacement therapy	NCT03729362	III
				Miglustat	Pharmacological chaperones	NCT04808505	III
				Duvoglustat	Pharmacological chaperones	NCT04327973	II
				SPK-3006	In vivo gene therapy vectors	NCT04093349	I/II
				Raav9-DES-Hgaa	In vivo gene therapy vectors	NCT02240407	I/II
				AT845	In vivo gene therapy vectors	NCT04174105	I/II
	Danon disease	LAMP2	LAMP2	RP-A501	In vivo gene therapy vectors	NCT03882437	I
Sphingolipidoses	Gaucher disease	GBA1	Glucocerebrosidase	Venglustat	Pharmacological chaperones	NCT02843035	II
				Afegostat	Pharmacological chaperones	NCT00813865 NCT00446550 NCT00433147	II
				Ambroxol	Pharmacological chaperones	NCT03950050 NCT04388969	II
				Miglustat	Substrate reduction therapy	[331]	Appd
				Eliglustat	Substrate reduction therapy	https://doi.org/10.1016/j.ymgme.2014.12.058	Appd
				Imglucerase	Enzyme replacement therapy	[332]	Appd
				Velaglucerase	Enzyme replacement therapy	[332]	Appd
				Taliglucerase α	Enzyme replacement therapy	[332]	Appd
				AVR-RD-02	Ex vivo gene therapy	NCT04145037	I/II
	Niemann–Pick type C disease	NPC1/NPC2	Sphingomyelinase	Miglustat	Substrate reduction therapy	[333]	Appd
				Arimoclomol	Inductor HSP70 synthesis	NCT02612129	III
				Ostat	Histone deacetylase inhibitor	NCT02124083	I/II
	Fabry disease	GLA	α-galactosidase A	Migalastat	Pharmacological chaperones	[334]	Appd
				Lucerastat	Substrate reduction therapy	NCT03425539	III
				Venglustat	Substrate reduction therapy	NCT02489344	II
				Agalsidase α	Enzyme replacement therapy	[335]	Appd
				Agalsidase β	Enzyme replacement therapy	NCT03018730	Appd
				PRX-102	Enzyme replacement therapy	NCT02795676	II
				AVR-RD-01	Ex vivo gene therapy	NCT03454893	Appd
				FLT190	In vivo gene therapy vectors	NCT04040049	I/II
				4D-310	In vivo gene therapy vectors	NCT04519749	I/II
				ST920	In vivo gene therapy vectors	NCT04046224	I/II
Neuronal Ceroid Lipofuscinoses	NCL II	CLN2	Tripeptidyl peptidase 1	AAVrh.10CUhCLN2	In vivo gene therapy vectors	NCT01414985	I/II
	NCL III	CLN3	Battenin	AT-GTX-502	In vivo gene therapy vectors	NCT03770572	I/II
	NCL VI	CLN6	CLN6 transmembrane ER protein	AT-GTX-501	In vivo gene therapy vectors	NCT02725580	I/II
Glycoproteinoses	α-mannosidosis	MAN2B1	α-mannosidase	DUOC-01	Intrathecal administration of cell therapy	NCT02254863	I

**Table 2 biology-13-00034-t002:** Overview of positive impacts of chemical autophagy inducers in models of lysosomal storage disorders.

Disease	Affected Enzyme	Product	Strategy	Ref.	Effects
Pompe disease	α-glycosidase	Rapamycin	Inhibition mTORC1.	[336]	Improved autophagic flux in patient myotubes.
Gaucher disease	Glucocerebrosidase			[337]	Enhancement of lifespan and locomotor activity in a Drosophila model.
Niemann–Pick type C disease	Sphingomyelinase			[338]	Restoration of autophagic flux and enhancement of cellular viability.
		Carbamazepine	Decrease in inositol and IP3 levels.	[338]	
		Verapamil	Blocking of L-type Ca^2+^ channel resulting in a decrease in cytosolic Ca^2+^ levels.	[338]	
		Trehalose	TFEB activation/SLC2A glucose transporters inhibition.	[338]	
		Lithium	Inhibition of IMPase leading to a decrease in inositol and IP3 levels.	[228]	
NCL III	Battenin			[339]	
		L-690,330		[339]	
		Trehalose	TFEB activation/SLC2A glucose transporters inhibition.	[340]	Mitigation of neuropathology and lifespan expansion.

## 5. Conclusions and Future Perspectives

The crucial role of lysosomes is breaking down macromolecules in eukaryotic cells. Defects in lysosomes’ function can lead to various human diseases with diverse causes. About 70 monogenic diseases have been linked to lysosomal impairment. Therefore, the scientific community is making significant efforts to explore potential therapeutic approaches for patients. These efforts focus on optimizing strategies not only to extend the lifespan but also to enhance the overall quality of life for patients. However, several challenges require thoughtful consideration. These include reducing the costs associated with the treatment and monitoring of chronic patients, improving therapeutic protocols, and introducing combination therapies and personalized medicine approaches. These improvements are vital in addressing the specific clinical issues faced by patients afflicted with complex multisystemic and heterogeneous disorders.

On the other hand, lysosomes are also implicated in several physiological processes such as autophagy, nutrient sensing, and intracellular signaling. In this context, for example, in the case of Pompe disease, the inhibition of autophagy affects the trafficking of the recombinant enzyme utilized in ERT [341]. Therefore, it is plausible to consider that enhancing the functionality of autophagic pathways could lead to an enhancement in the delivery of the therapeutic enzyme to lysosomes not only in the context of Pompe disease but also in the case of other LSDs. It is not surprising that other potential therapeutic targets may be identified by the precise characterization of the molecular pathways implied in the development of LSDs, and this will open new opportunities for effective treatment in the context of LSDs.

## Figures and Tables

**Figure 1 biology-13-00034-f001:**
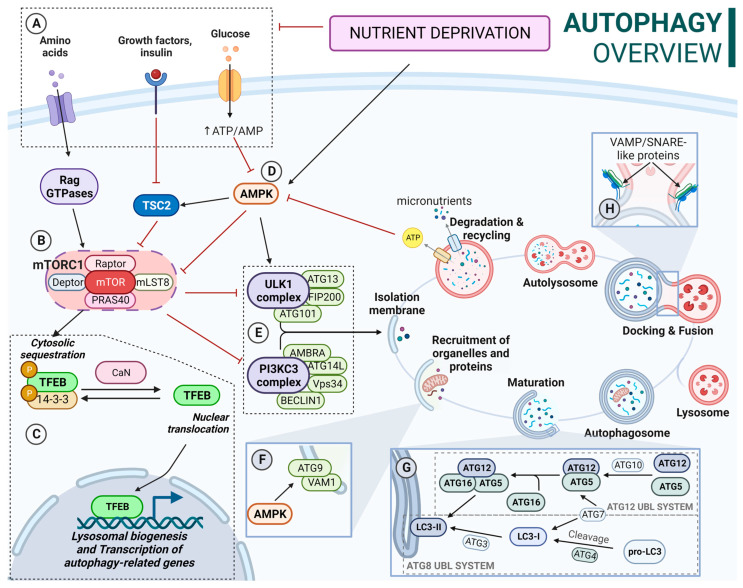
(**A**) Nutrient-availability conditions: Increased amino acid availability activates Rag GTPases. Active Rag GTPases recruit mTORC1 to the lysosomal membrane and activate it. TSC2 is inactivated by insulin and IGF1. AMPK is inactivated by increased ATP/AMP ratio. (**B**) mTORC1 influences several processes including TFEB location and ULK complex and PI3KCIII complex activity. mTORC1 is inhibited by TCS2 and AMPK. (**C**) TFEB is phosphorylated by mTORC1 in several residues; phosphorylation of Ser-211 promotes its binding to 14-3-3 proteins, resulting in a cytosolic location. Calcineurin mediates TFEB dephosphorylation, allowing its translocation to the nucleus. Calcineurin activation is a Ca^2+^-dependent process. In this context, after mTORC1 release from the lysosomal membrane, some of the Ca^2+^ lysosomal deposits are released to the cytosol, contributing to calcineurin activation. (**D**) Active AMPK inhibits mTOCRC1; activates TSC2; phosphorylates ULK1, promoting ULK1 complex activation; and phosphorylates ATG9, promoting its mobilization towards autophagosome formation sites. (**E**) Active ULK complex recruits the PI3KCIII complex, triggering the formation of the isolation membrane (IM). (**F**) Transmembrane proteins VMP1 and ATG9 participate in the recruitment of membranes to expand the IM. (**G**) ATG12 UBL: ATG5-ATG12 conjugation takes place by ATG7 (E1) and ATG3 (E2) sequential action. Then ATG12-ATG5 conjugate binds to ATG16L, forming the ATG12-ATG5-ATG16L complex. ATG8 UBL (LC3B shown): Inactive LC3 is activated when cleaved by ATG4. LC3-I is conjugated with PE to LC3-II (lipidated form) on the elongating autophagosome membrane. This is achieved by the sequential action of ATG7 (E1), ATG10 (E2), and ATG12-ATG5-ATG16L (E3). (**H**) SNARE proteins such as STX17or SNAP29 interact with R-SNAREs like VAMP7 or VAMP8, leading to fusion. In addition, STX17-SNAP29 complex stabilization by ATG14L is required for successful fusion.

**Figure 2 biology-13-00034-f002:**
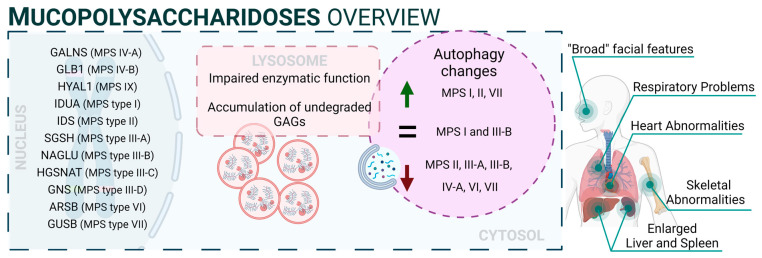
Mucopolysaccharidoses overview.

**Figure 3 biology-13-00034-f003:**
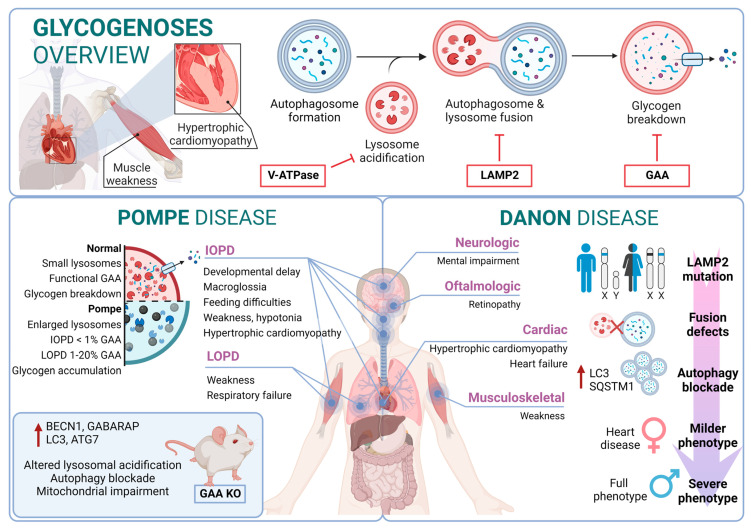
Glycogenoses overview. Pompe disease and Danon disease.

**Figure 4 biology-13-00034-f004:**
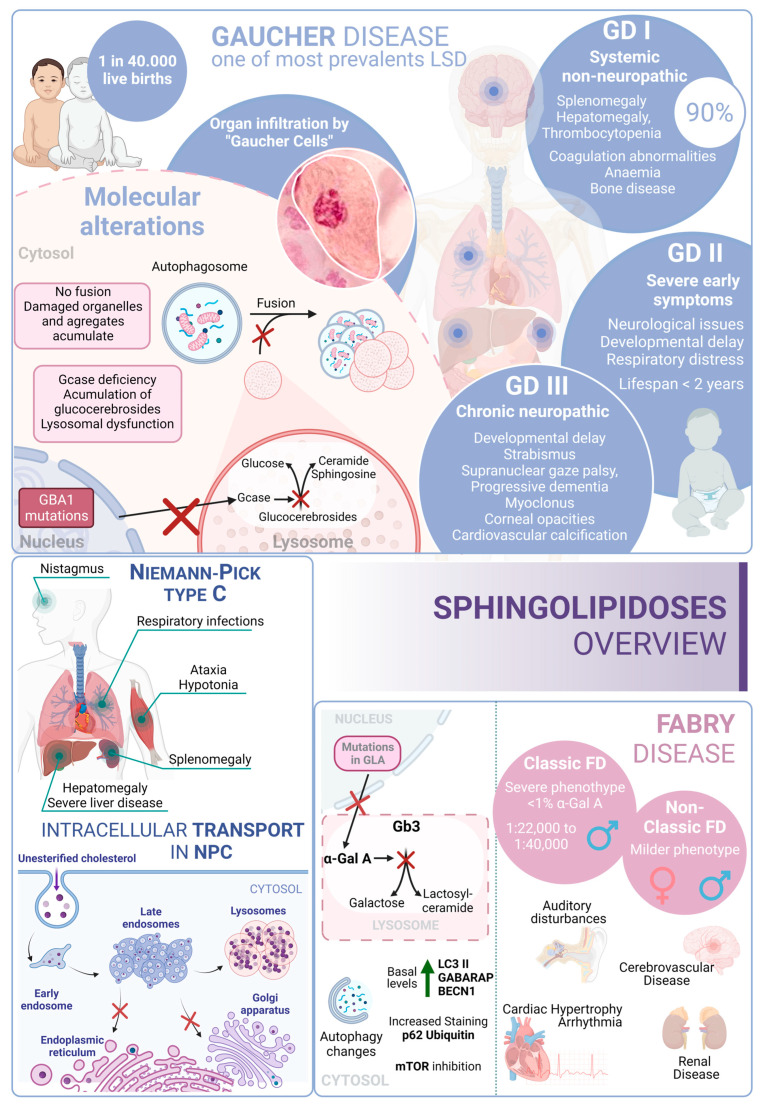
Sphingolipidoses overview. Gaucher disease, Niemann–Pick type C disease, and Fabry disease. GD I, II and III correspond to the three different presentations of Gaucher Disease.

**Figure 5 biology-13-00034-f005:**
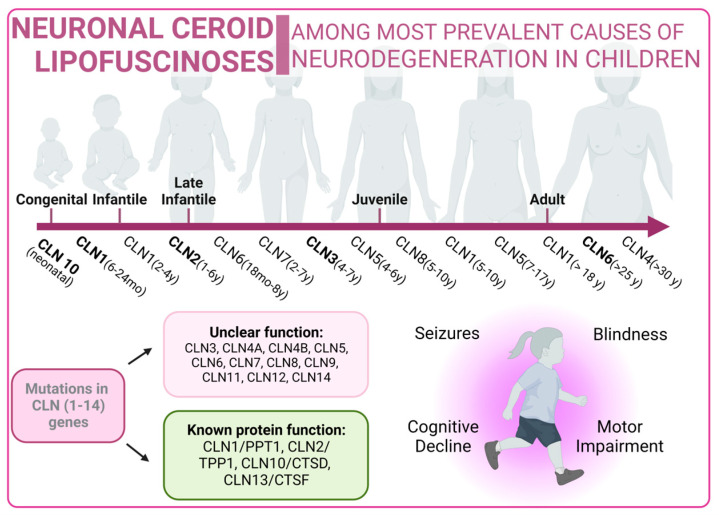
Neuronal Ceroid Lipofuscinoses overview. Classical forms of the disease are marked in bold type.

## Data Availability

No new data were created or analyzed in this study. Data sharing is not applicable to this article.

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
