# Peer review of "Lysosomal Dysfunction: Connecting the Dots in the Landscape of Human Diseases"

_biology, 2024, doi:10.3390/biology13010034_

Round 1

Reviewer 1 Report

Comments and Suggestions for Authors

The review by Uribe-Carretero et al describes molecular/cellular mechanisms of Lysosomal Storage Diseases. the manuscript is long and some of the information could be tabulated. It would shorten the text. the table could summarize mechanisms and potential therapies targeting the defect.

Additional figures could make the mechanisms clear for each disease. 

A list of diseases is not extensive. What about alpha and beta mannosidosis?

The list of references is not extensive: The authors could consider referencing: DOI: 10.3390/jcm9082596.

Comments on the Quality of English Language

Supplementary file: Please replace 'Morchio A' with 'Morquio A'

Author Response

Dear reviewer:

In first place, we o express our gratitude for your valuable suggestions. Below, we proceed with our response to your comments point by point:

We have done our best to shorten some parts of the manuscript. We have included a new figure in the autophagy section, which has allowed us to summarize and clarify that point.

We believe that the figures we have incorporated provide a simplified overview of mechanisms along with the main clinical manifestations of each of them.

Our objective with this article was to review the most common disorders within this group of diseases, so initially, we did not considered to include the diseases you propose. However, we have added a new section (3.5) for the group of pathologies you suggested.

We have updated the manuscript to include references to all the articles reviewed to elaborate the tables. We have also added the suggested reference, the references fot the new group included, along with all the articles and clinical trials for all the diseases addressed in this review.
Our best regards.

Reviewer 2 Report

Comments and Suggestions for Authors

The authors Uribe-Carretero et al., have submitted a review article entitled "Lysosomal Dysfunction: Connecting the Dots in the Landscape of Human Diseases". The authors have written a comprehensive manuscript that highlights the role of autophagy and various pharmacological interventions in treating lysosomal storage disorders. A few changes, if made, will help to enhance the impact of the article - 

On lines 67 - 70, the authors mention that the hallmarks of health have been categorized into three different categories. However, the authors did not mention or elaborate on these categories. It would also be helpful to mention how autophagy plays a role in each category. This would help to link why autophagy is a mechanism for promoting health and longevity.

The lines 91 - 100 do not fit in their current location. The authors could move them to the text of lines 62 - 66.

The scope of the review on lines 101 - 102 needs to include the point that the authors will focus on the role of autophagy and pharmacological interventions in lysosomal storage diseases.

As the authors are describing autophagy as a separate section, it would be helpful to include a Figure describing autophagy with all the protein level details mentioned in the text on autophagy.

The title of section 3.1 could be revised to be more in line with the titles of the other sub-sections (3.2, 3.3, etc).

Figure 2, similar to Figure 3,  could have a title like Glycogenoses overview as part of the figure.

The section on pharmacological modulation could be revised to include a description of which treatment therapy is most effective in which type of LSD. Also, it would be helpful if the authors also described if any of these pharmacological interventions target autophagy while treating LSDs.

Comments on the Quality of English Language

None

Author Response

Dear reviewer,

Firstly, we would like to express our gratitude for all your helpful suggestions. Below, we proceed with our response to your comments point by point:

  1. We have added the categories introduced in lines 67-70. We agree with your input in this regard, and it is now better covered and connected with the contribution of autophagy to health and longevity.

  2. We have moved lines 91-100 to the suggested part of the manuscript. We agree that this change improves the connection.

  3. We have updated the scope of the manuscript to clarify that we will focus on autophagy and pharmacological approaches.

  4. We have created a new figure to display most of the autophagy mechanisms and connected pathways from the autophagy section. We believe this will clarify this part of the manuscript. We appreciate this suggestion as it has a very positive impact on the quality of the manuscript.

  5. We have updated the section titles as well as the figures to maintain consistency throughout.

  6. We have revised the pharmacology section to include two tables. One includes all the treatments for each disease, including approved ones as well as those under clinical trials. The other includes autophagy modulators studied in disease models, along with their mechanisms and effects on the mentioned models.

Thank you for your time and your suggestions. We wish you a Merry Christmas and a Happy New Year.

Round 2

Reviewer 1 Report

Comments and Suggestions for Authors

The authors addressed the reviewers' comments sufficiently